# Assessing the Educational Potential of Geosites: Introducing a Method Using Inquiry-Based Learning

Emil Drápela

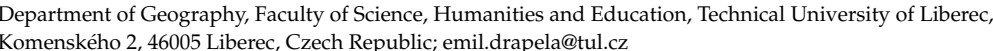

Department of Geography, Faculty of Science, Humanities and Education, Technical University of Liberec, Komenského 2, 46005 Liberec, Czech Republic; emil.drapela@tul.cz

**Abstract:** Geosites are suitable locations for field teaching of Earth sciences. However, their educational potential does not always correlate with the scientific significance of geosites, as for educational purposes, the visibility and comprehensibility of the phenomenon are much more important. The educational potential also depends on the target group, as a location suitable for the education of adults may not be suitable for the education of younger pupils. The article describes an experiment in which a method of assessing the educational potential of geosites was developed based on the analysis of the outputs of inquiry-based learning tasks during field teaching on geosites. The method is based on the gradual implementation and evaluation of the inquiry-based learning program for different categories of target groups, proceeding from more experienced and older to less experienced and younger participants. Although the method is relatively time-consuming, it provides very accurate results that can be applied to different target groups. The use of this method can help schools, institutions implementing extracurricular education programs, and geoparks to identify correctly suitable geosites.

**Keywords:** geoeducation; inquiry-based learning; geographic education; geosite assessment; field training; science popularization; Ralsko National Geopark

## 1. Introduction

The development of geotourism in the last two decades is accompanied by the need to create a tool for geosite assessment [1]. It is necessary to evaluate how valuable the selected geosites are from a scientific, touristic, educational, aesthetic, and cultural point of view, etc. On the basis of this assessment, localities are selected that are promoted and made available for geotourism. There are many assessment methods used. Their overview is given, for example, in the articles by Strba et al. [2] or Brilha [3], but many others have been created in recent years as this topic has been experiencing great development [4–10]. Many authors of these methodologies then admit that while certain parameters used by their methods can be defined quite precisely, others are relatively difficult to determine, and the determination of their value is subjectively influenced [2,11,12].

Among such parameters, which are relatively more difficult to define, is the educational value of the geosite. If we only numerically express the number of phenomena that can be demonstrated at a given location (or other numerical expressions of the "objective criterion"), we will obtain an assessment that does not correspond to reality, as the educational value is rather based on the understanding of the phenomenon when visiting the location [13–15]. Here, we get to the crux of the problem—what is understandable for an adult is not understandable for a 10-year-old child and vice versa. A university student of geology will appreciate a different geosite than an ordinary visitor on a guided walk in the geopark. So, how can the educational value of a geosite, or rather the educational potential of a geosite, be measured?

First, it is necessary to come to terms with the fact that the educational potential of a geosite is not a universal value but is dependent on the target group. The author of

this article often creates educational programs for different age groups, from children in kindergarten to groups of seniors, and for groups of various education and motivations, from ordinary tourists, through enthusiastic laymen, to groups of professionals. The selection of suitable geosites for an excursion depends significantly on the target group, as a number of geosites that are very interesting for professionals will not interest children or ordinary tourists, and vice versa [15]. From the point of view of education, the key element is the comprehensibility of the phenomenon by the given target group, when ideally we want our audience to be able to imagine the geological and physical–geographical phenomena that created what they see in front of them. Based on this experience, they should then be able to explain this phenomenon themselves—this is the best check that they have really understood the explanation.

Second, if we accept the fact that the educational potential is dependent on the target group, a method needs to be developed to find out which topic in which location is suitable for which target group. For this purpose, in this article below, experimental verification is proposed, which is implemented on variously experienced target groups using inquiry-based learning. Before this method is described, the term needs to be explained.

Inquiry-based learning is a pedagogical method in which the activity of the teacher and the pupil is focused on the development of knowledge, skills, and attitudes based on active and relatively independent recognition of reality by the pupil, which he himself learns to discover [16]. It includes both the activity of the teacher, who creates a teaching scenario, and the activity of the student—research through which he learns about the surrounding world [17]. The result of the student's research is a subjectively new discovery that is already known to society but is of great importance to the student because through it he will understand generally valid phenomena, which he will remember better thanks to the intensive experience [16]. The teacher (or georanger) has a role of a facilitator rather than a lecturer—he identifies the research question and provides the student with the necessary materials so that the student is able to come up with the answer himself [18].

The use of inquiry-based learning requires special training from educators, as it is a method that is more demanding than the implementation of ordinary frontal interpretation [19]. However, it is usually more fun and informative for students, because they themselves are involved in the process of acquiring information [20]. Research questions need to be carefully considered in relation to the age and knowledge of the student, as tasks that are too difficult demotivate the student, and tasks that are too light are boring [16,20]. If we use inquiry-based learning in geology and geography, we often work with the landscape around us. In such a case, it is necessary to consider whether the answer to the research question is sufficiently visible and understandable in the landscape [21]. That is, whether the chosen place has the right educational potential for the given target group. Therefore, if we experimentally verify with the help of successfully implemented inquiry-based learning that the educational potential for the selected target group exists here, we can take it as a fact. This principle is used by the method described below.

## 2. Materials and Methods

The method of assessing the educational potential of the geosite is based on an experiment in which an educational program using inquiry-based learning takes place, followed by verification using the evaluation of acquired knowledge during a regular educational program. The experiment uses the classification of potential target groups into categories according to the volume of relevant knowledge and intellectual abilities. This classification is shown in Table 1.

The division of target groups into categories is conducted because if the combination of the educational topic (selected phenomenon) and geosite does not work for a certain category, it can still be used for a higher category. Category 6 (professionals) can also appreciate a less aesthetically attractive location and a less visible phenomenon if both are interesting enough. However, geotourists coming for excursions and educational programs

in geoparks tend to be in categories 4–5 or (sometimes) 3. Lower categories need to be addressed only if the subject is engaged in the education of children.

**Table 1.** Classification of potential target groups according to the volume of relevant knowledge and intellectual abilities.

| Category | An Example of a Target Group |
|:---:|:---:|
| 1 | Kindergarten children |
| 2 | Primary school children |
| 3 | Secondary school children, ordinary tourist |
| 4 | High school student, a layperson with a mild interest in geology (ordinary geotourist) |
| 5 | University student (relevant field), a layperson with a deeper interest in geology (member of a natural science association, reader of popular geological literature, etc.) |
| 6 | Professional |

The categories of target groups are used in the experimental verification of the educational potential of the geosite, an overview of which is shown in Figure 1. First, the educator selects the geosite and the topic of the educational program in detail, which he considers appropriate for the selected target group. Emphasis is placed not only on the objective parameters of the geosite (scientific value, etc.) but also on the subjective effect of the geosite (aesthetic value, etc.) and especially the visibility of the topic or phenomenon in the terrain (comprehensibility based on sensory perceptions in the terrain, etc.). If the educator is convinced that the program proposed should be adequate for the selected category of target groups, it is possible to proceed with its implementation.

**Figure 1.** Overview of the experimental verification of the educational potential of the geosite using the inquiry-based learning program.

As part of the implementation, a program scenario is first created according to the generally valid principles of good interpretation (for more information, see [22–26]). Furthermore, it is necessary to prepare all the tools and materials that will be needed for the inquiry-based activity. This usually forms only part of the overall program, but it should be a pivotal part. The educator should first introduce the topic or phenomenon but should

not reveal too much about it. This introduction should work more as a motivation when listeners are interested in the presentation and want to learn more about the topic. In the further process, the educator can draw attention to various "clues" that will later help the participants to solve the assignment of the inquiry-based activity. When the field trip arrives at the geosite that is the main goal of the day, it is time for the educator to explain the assignment of the inquiry-based task. He then gives the participants a reasonable amount of time to try to get it right.

The evaluation of acquired knowledge then takes place in two steps. Immediately after the expiration of the time to complete the inquiry-based task, the percentage of participants who were able to find the correct answer or successfully complete the task is recorded. The educator notes this number and then explains the correct answer. The rest of the program follows, which should answer all of the participants' questions or ambiguities regarding the topic or phenomenon. At the end of the excursion, it is assumed that all participants already understand everything, so it is the right time to carry out the second phase of the evaluation with the help of a short test. The test will show whether the participants themselves could simply explain the main ideas that were the central theme of the excursion. So that the participants do not have to write for a long time, it is advisable to use the sentence completion method. Participants submit their answers to the educator, who will evaluate them only after the excursion. The indicator is again the proportion of correct or almost correct answers.

The result of the evaluation of the inquiry-based learning activity is therefore two percentage data, indicating the degree of success (a) in fulfilling the assigned activity and (b) in understanding the given topic or phenomenon. Both of these values should ideally be higher than 70% (why this particular value is explained in the Discussion chapter). If the result of an inquiry-based activity does not exceed the threshold of 70%, its assignment was too difficult. If the result of the final test does not exceed this limit, there was probably a bad performance by the educator or the creation of the excursion scenario (or other reasons that can be considered in retrospect). However, if both results do not reach the 70% threshold, the program is unsuitable for the selected category of target groups. If the success rate in both indicators exceeds 70%, it is possible to move down one level in the table of categories of target groups. If, for example, the program is successfully tested by a group of university students (category 5), it is possible to subsequently try it with a group of high school students or on an excursion organized by a geopark (category 4).

Given that the above-mentioned description is only general, the following chapter will present its application to one selected locality in the Czech Republic, namely Velký Jelení vrch in Ralsko National Geopark (see Figure 2), with the theme "geological evolution of the surrounding landscape". Since the study involved work with people, it should also be noted that the research was in accordance with the Code of Ethics of the Technical University of Liberec, based on the valid laws of the Czech Republic and European Union regulations. The research met the standards usual in the social sciences, and all participants took part in the research voluntarily.

*Description of the Experiment*

An excursion scenario was compiled for the selected geosite (Velký Jelení vrch) and topic (geological evolution of the surrounding landscape), including an element of inquiry-based learning. An overview of this scenario, including the design of the experiment, is presented in Table 2. A route was chosen for the excursion, which first introduces the participants to the three main types of rocks found in the area. The first of them are sandstones of the Cretaceous age, which are the remains of a sea flood 90–65 million years ago [27]. Locally unique are the polzenites (a collective name for olivine and melilite rocks with nepheline), igneous rocks that filled the cracks in the sandstone and formed as a result of Alpine folding about 75 million years ago [28]. The third type of rocks are basaltoids, which are related to volcanic activity in the Tertiary, which is again related to the progressing Alpine orogeny [29]. These three types of rocks can be distinguished from

each other even by a preschool child, because sandstone has an ocher color and visible sand grains, polzenite is light gray and forms plate-like bodies, while basaltoids are dark gray, very heavy, and in some places form columnar cleavage. Each of these rocks then creates a different type of relief: sandstones create flat mesas, polzenites narrow steep ridges, and basalts lonely high hills. Understanding the relationship between geology and geomorphology, along with other interesting facts about the human use of these resources in the past, is the central theme of this excursion.

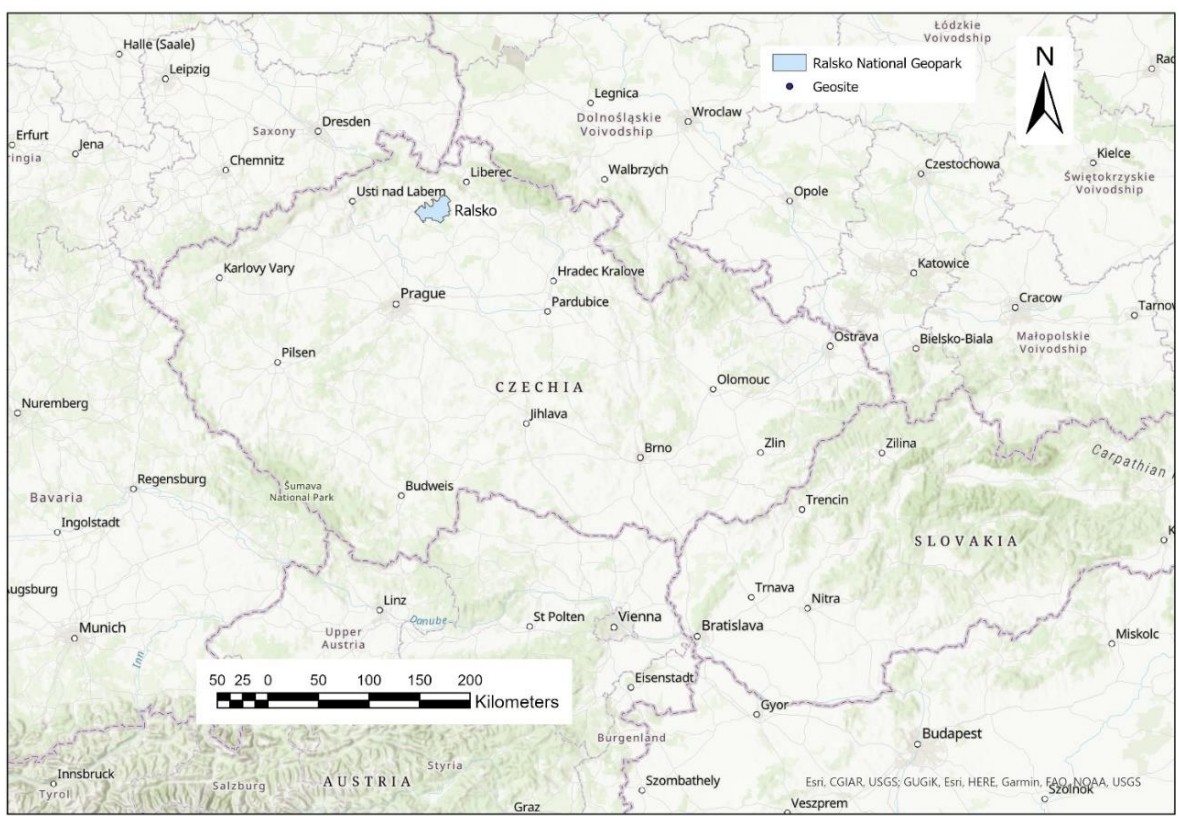

**Figure 2.** Location of Ralsko National Geopark and geosite Velký Jelení vrch within the Czech Republic.

**Table 2.** Experimental design data overview.

| Main topic | Geological Evolution of the Surrounding Landscape |
| --- | --- |
| Place | Velký Jelení vrch, the northern part of the Ralsko National Geopark, the northern edge of the Bohemian Cretaceous Basin. |
| Target group | Category 5 and lower. |
| Excursion design | The starting point is the village of Hamr na Jezeře, from where the 12 km long circuit starts. During the first eight kilometers, participants will encounter the three main types of rocks in the area and see the specific relief shapes they create. However, the guide does not explain their genesis. An inquiry-based activity follows. In the last third of the journey, the knowledge gained during the inquiry-based activity is repeated so that even individuals who could not successfully complete it can understand its message. |

**Table 2.** *Cont.*

| Main topic | Geological Evolution of the Surrounding Landscape |
|---|---|
| Inquiry-based activity setting | We are located on a viewpoint from where there is a nice view of the surrounding hills. Try to create a simple panoramic sketch in which you name these hills and write down what kind of rocks they are made of. A geological map will help you with this. Next, try to find out in which order these rocks were formed and how they affect the shape of the hills they form. Finally, try to create a short (5–7 sentences) "story of the evolution of the surrounding landscape", in which you simply explain how what we see was created. You can use the internet on your mobile for this. |
| Necessary tools for inquiry-based activity | A section of the geological map of the area with a good topographic background. |
| Assessing the success of an inquiry-based activity | The participant should be able to identify some examples of hills made up of all three main rock types found in the area. Furthermore, he should be able to define how their shapes differ, in what order they were created, and during which processes. If he gives these data correctly or almost correctly (with some small errors), the result of his activity is evaluated as successful. |
| Short final test setting | A text of 10 sentences in length, in which some terms or parts of the explanation are omitted, which the participants have to complete. In total, participants have to complete 11 words or parts of sentences. The test takes about 5 min to complete, and participants are not allowed to cooperate. |
| Assessing the success of a short final test | If the test is completed correctly or with one error, the result is considered successful. |

The excursion is 12.2 km long; due to the hilly terrain, the walk will take about four and a half hours (see Figure 3). Stopping at geosites, interpretation by the guide and inquiry-based activity will take another 4.5 h, so overall the excursion is a full-day trip. The starting point is the nearby village of Hamr na Jezeře, from where the excursion goes to the ruins of Děvín Castle. During the journey, the contrast between the vegetation growing on the sandstone bedrock (poor pine forest with blueberries in the undergrowth) and on the mineral-rich volcanic rocks (beech forest with a rich shrub and herb layer) is clearly visible. Furthermore, the participants are introduced to two local rocks: sandstone and polzenite. At the Děvín geosite, they can admire both the polzenite vein and the interesting iron incrustations in the sandstone. The route continues to Schachtenstein, copying the line of the polzenite vein. It was mined at Schachtenstein in the Middle Ages, creating an interesting mining monument. From the top of Schachtenstein, there are views of the surrounding sandstone mesas, especially of the nearby Široký kámen. Another geosite is Kozí hřbety, a very narrow hill whose core is a vein of polzenite. In one place, the path goes on the top of a roughly two-meter-wide vertical cliff, which consists of polzenite dissected from the surrounding sandstone. For most visitors, walking through this section is a great experience.

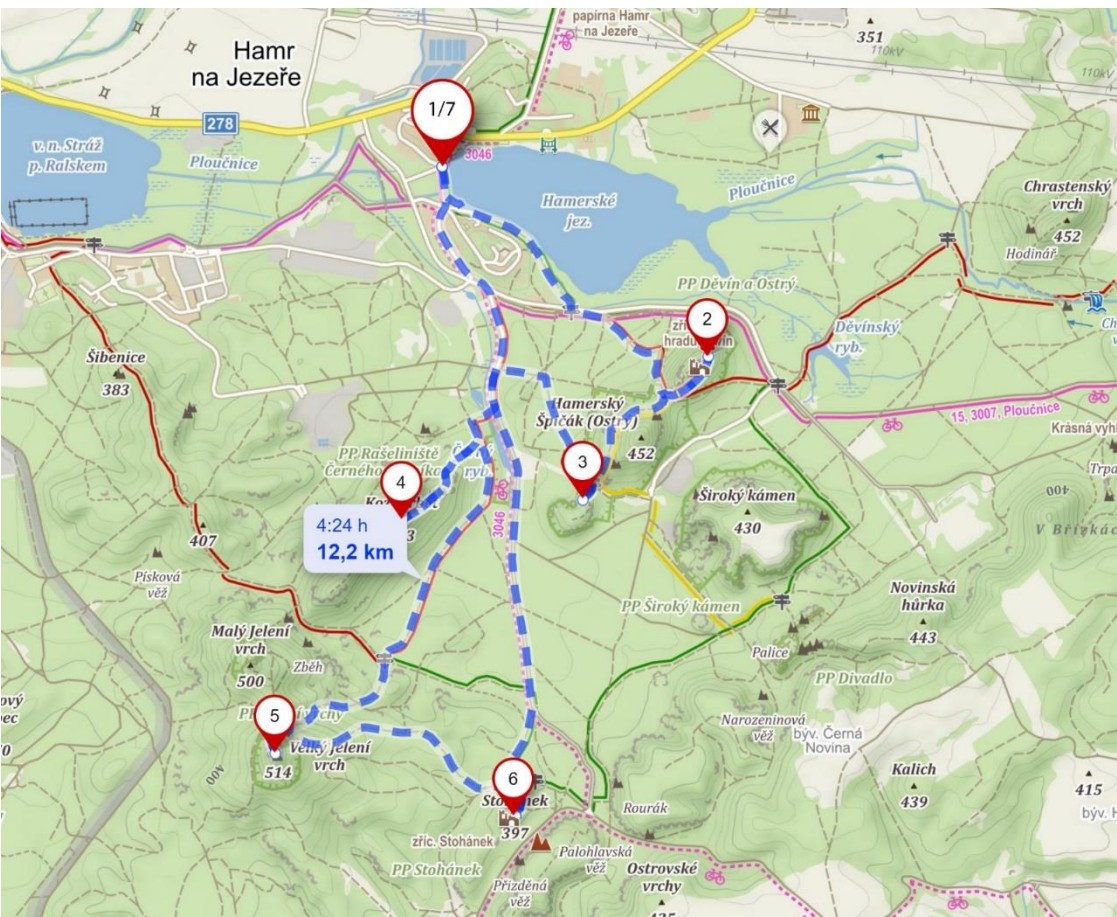

**Figure 3.** Excursion route map. The marked points are the main visited geosites: 2—Děvín, 3—Schachtenstein, 4—Kozí hřbety, 5—Velký Jelení vrch, 6—Stohánek.

After about two-thirds of the excursion's length, it comes to the place that is its highlight: Velký Jelení vrch. It is a peaked rock on a basalt vein, from which there is an almost circular view. In good weather, you can see tens of kilometers away, but during the excursion, only the immediate surroundings, which are clearly visible, are used. An inquiry-based activity and its assessment take place on this geosite (see Table 2). The time spent on this activity depends on how long the group needs to rest (or time for a snack), usually between 20 and 30 min. The guide will first explain the activity (see Table 2), answer any questions, hand out geological maps (as necessary tools for the activity), and then be available for any consultation. However, he does not reveal or suggest the correct solution to the participants. The geological map that the participants will receive is shown in Figure 4. For the purposes of this article, pins with referenced geosites have been added to them. On the other hand, for clarity, an extensive legend available in the source map application of the Czech Geological Service [30] is missing. Participants have this legend printed out so that they can use it to successfully complete the task. After the time limit has expired, the guide first collects the results of the participants' efforts and then invites them to answer the questions that he gave at the beginning. The participants will thus put together the correct solution together, or the guide will correct their answers. The approach of the guide and his communication with the participants must be adapted to the target group; there is a big difference between, for example, a school team and a group consisting of parents with children.

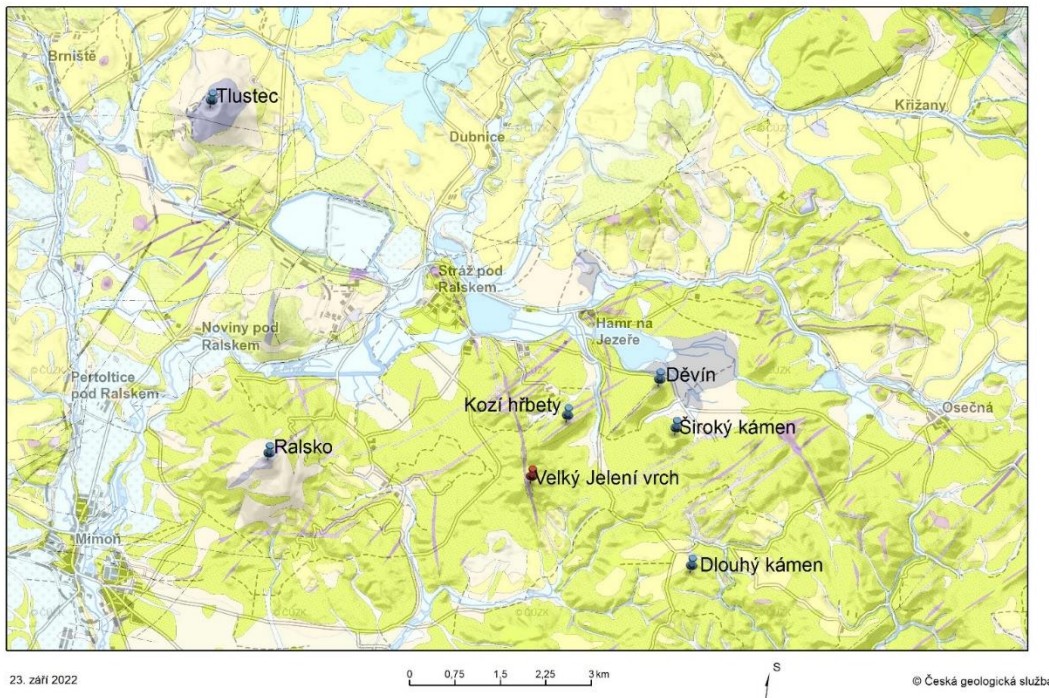

**Figure 4.** Simplified geological map of the vicinity of Velký Jelení vrch with the marking of some important geosites. Source: Czech Geological Survey [30]. Legend is available at online map application: https://mapy.geology.cz/geocr50/ (accessed on 15 October 2022).

In order to successfully complete the task, it is necessary to combine information from the course of the excursion so far (experience in the field, knowledge of the three main types of rocks in the area, or the fact that leafy trees indicate volcanic rocks in the subsoil), visual perception of surroundings, geological maps, and possible information on the Internet (e.g., in case some terms from the legend are unknown to the participants, typically geological periods). Parts of the view are shown in Figure 5, and the three main rock types found in the described area are shown in Figure 6. The researcher will only evaluate the successful completion of the task after the excursion. Participants themselves will know if their answers were correct and will not feel embarrassed if they fail. The guide should shift attention from the completed papers to the surrounding view and invite the participants to tell him what they have discovered. The result of this process should be the delivery of the entire main message of the excursion to the participants. After that, they should be able to reproduce it themselves.

To ensure that the journey back to the starting point is not without a visit to other attractions, the excursion includes a visit to the Stohánek geosite, which is a small mesa with nice views of the surroundings and the remains of a guard castle from the Middle Ages at the top. Due to the visible damage to the site by tourism, the main topic here is the sustainability of tourism and the threat of damage to geosites due to overtourism [31,32]. Unfortunately, in the Czech Republic, a number of similar sites in sandstones are threatened not only by the constantly growing number of visitors but also by their inappropriate behavior [33]. On guided excursions, it is always advisable to point out this danger and act as a precaution, especially when it comes to school groups.

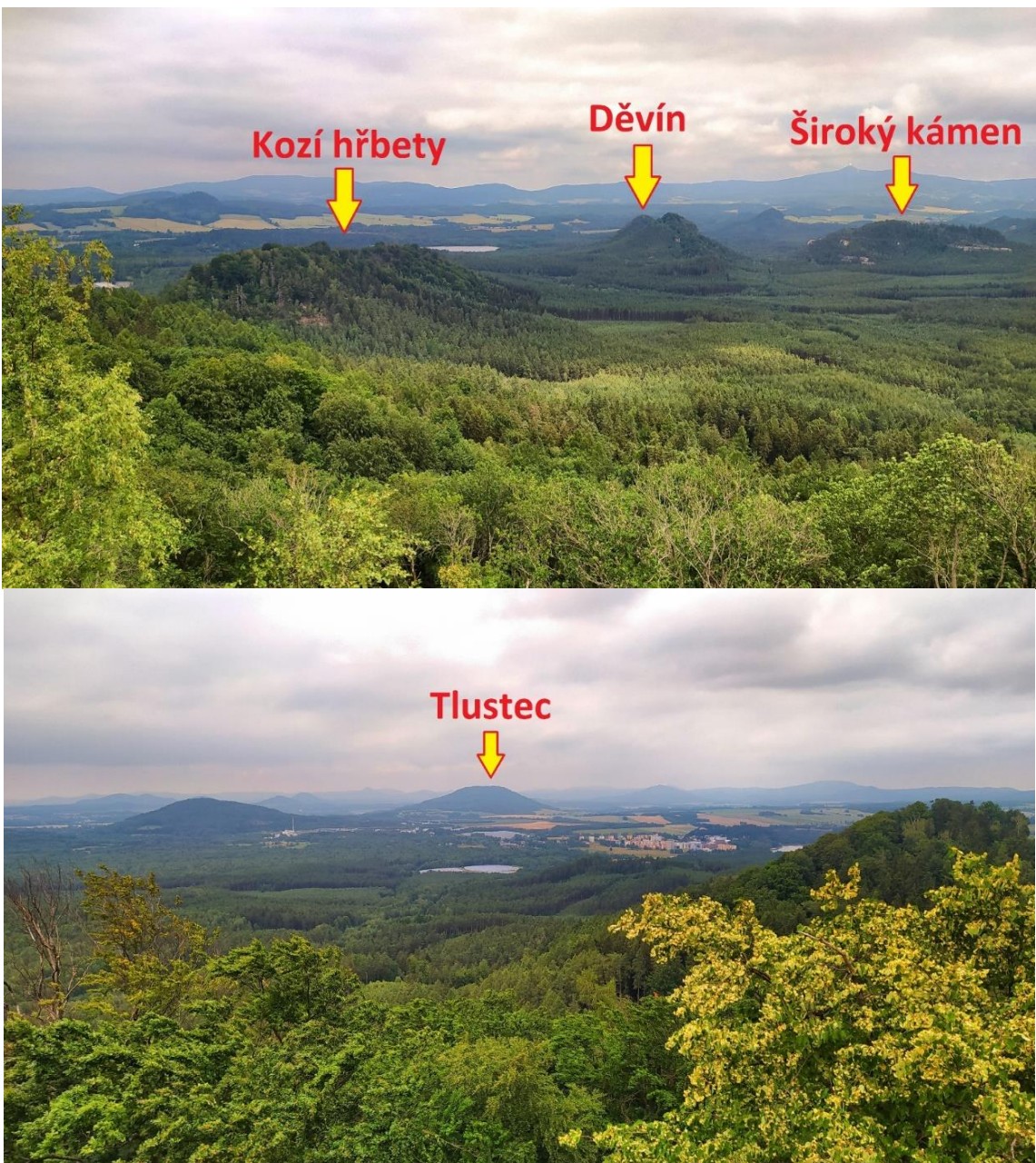

**Figure 5.** Parts of the view from Velký Jelení vrch illustrating the differences between solitary basalt hills such as Tlustec, sandstone mesas such as Široký kámen, and narrow polzenite ridges such as Děvín and Kozí hřbety (Kozí hřbety, however, are tilted perpendicular to the axis of view in the photo). Photos by author.

At the end of the excursion, there is a final test, which should take participants about 5 min. This is a short text summarizing the main findings from the excursion. However, a total of 10 words or parts of sentences are left out in this text, which the participants have to complete themselves. In this way, we test how much information the participants actually remembered. It is important that the participants do not feel that they are being "tested", especially in a situation where they are tourists who voluntarily came on a walk organized by the geopark. In such a case, it is necessary to work with humor and exaggeration and not to force the participants to take the test, but to motivate them (e.g., motivation works well for families with children when "those who complete the test well will get something sweet", etc.).

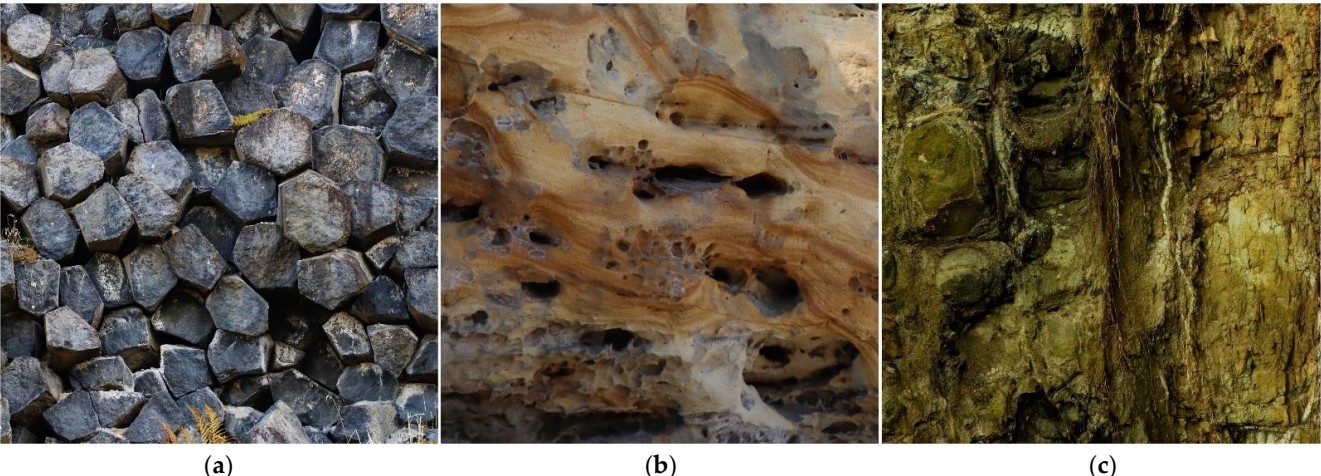

|  (a)  |  (b)  |  (c)  |

**Figure 6.** The three main rock types found in the described area: basalt (**a**), sandstone (**b**), and polzenite (**c**). Photos by author.

The following text is used in this excursion: "Today's excursion passed through the northern part of the Ralsko National Geopark. It belongs to the Ralská pahorkatina upland, in which there are three types of hills depending on the geological bedrock. The mesas are made of a rock called (sandstone) and were formed in such a way that (the originally flat relief was modeled by erosion). The narrow ridges forming long lines have a rock called (polzenite) at their core and are so steep because (polzenite is a more resistant rock and the hot solutions during volcanic activity also solidify the surrounding sandstone). The highest hills are made of (basalt) which sometimes creates long columns. It was formed from (hot magma), which mostly solidified below the surface. But in some cases it reached the surface—for example, a nearby hill (Ralsko) is a former volcano. The surrounding relief developed in such a way that first in the (Cretaceous) period everything was covered by a shallow sea, which left behind powerful layers (sandstones). These later cracked due to ongoing (Alpine folding) and volcanic rocks penetrated these cracks. The last manifestations of volcanism in the surrounding area is about 15 million years old; since then the surrounding relief has been subject to (erosion), which has shaped it into its current forms."

The above-mentioned excursion was carried out in 2021 and 2022 with participants of target group categories 5, 4, and 3. In category 5, there were two groups of university students studying geography teaching, with a total number of 86 participants. In category 4, there were two groups of gymnasium students ($N = 54$) and one group of people on a regular geotourism excursion ($N = 37$). In category 3, there were two groups of secondary school pupils ($N = 52$), after which the experiment was terminated.

## 3. Results

The results of experimental testing are shown in Table 3. For both groups of university students, the success rate was around 90% (that is, well above the 70% mark), so we proceeded to category 4 testing. Gymnasium students and visitors to the geotourism excursion again achieved a result higher than 70%, but slightly lower than university students. Visitors to the geotourism excursion as the only group achieved a worse result in the final test than in the inquiry-based activity, which is objectively more difficult. This may be due to greater distraction during the excursion, when parents had to take care of children, etc. Then, a certain part of the information could have escaped them. During the testing of secondary school students, a significantly lower success rate than the required 70% was achieved. The reaction to this was the grouping of pupils into threes and not pairs as before, but even then the success rate was too low. Therefore, the testing was terminated.

As a result, this educational program is suitable for target groups of categories 4 and 5, but not for the categories below.

**Table 3.** Results of experimental verification of the educational potential of geosite Velký Jelení vrch.

| Category | Number of Participants/Groups | The Inquiry-Based Activity Success Rate | The Final Test Success Rate |
|---|---|---|---|
| 5 | 42/21 | 90.47% | 95.23% |
|  | 44/22 | 86.36% | 90.90% |
| 4 | 28/14 | 78.57% | 85.71% |
|  | 26/13 | 69.23% | 84.61% |
|  | 37/13 | 84.61% | 76.92% |
| 3 | 28/14 | 42.86% | 57.14% |
|  | 24/8 | 50.00% | 62.50% |

Why did the success rate for category 3 drop significantly? Pupils confused both types of volcanic rocks, had a problem with reading the geological map and orientation in the map in general, and could not imagine the plate-like body of a volcanic vein, etc. The biggest problem then was the task of reconstructing geological history. In order for this program to work well even for this category, it would be necessary to simplify it even more (just sedimentary vs. volcanic rock). However, the question is whether there is a more suitable geosite in the vicinity for this simplified option.

The aim of this article is to present the methodology for evaluating the educational potential of geosites, which is presented in one specific program. However, to be able to trust this methodology, it needs to be tested in a larger number of locations. That also happened. In the years 2018–2022, the methodology was applied to a total of 46 educational programs (combination of geosite and topic), the result of which is a constantly growing database of potential targets of geotourism interest. Furthermore, certain similarities were found in the focus of the tested educational programs, which are shown in Table 4.

**Table 4.** Different types of educational programs according to the results of experimental testing using the method described above.

| Suitability for Categories | The Nature of the Educational Program | Suitable Geosite |
|---|---|---|
| 1 + 2 | Primarily playful form, very simple message. | Near the starting point, scientific value is not important. |
| 2 | A slightly professional program for children interested in nature. | A place where you can do some interesting activities (collecting minerals and fossils, panning, rock climbing, etc.). |
| 3 | The program is closely related to the curriculum currently being discussed at secondary school. | An attractive geosite, the focus of the program is often narrowly defined, so the potential of the geosite cannot always be used sufficiently. |
| 3 + 4 + 5 | A classic excursion for the general public, for whom no greater knowledge is assumed. | An attractive geosite where an interesting story can be presented. |
| 4 + 5 | An excursion aimed at a motivated geotourist or a student of a specialized school. | Geosite with a balanced ratio of attractiveness and professional interest. The main message of the excursion is more difficult to understand. |
| 5 (6) | A visit to a site of professional interest. | A geosite of high scientific value. |

## 4. Discussion

In many published works, the educational value of a geosite is derived from its scientific value or presented as a certain expression of the number of phenomena that can be shown on a geosite [7,34–36]. However, this does not correspond to the reality of the educational process, when education must be engaging with an interesting story, should affect emotions, be illustrative and comprehensible, and work in the field with the help of one's own experience [22–26]. Just as children's intellects develop during their adolescence, so do the means of best educating and influencing them. Moreover, although there are some very attractive geosites that can impress people of all ages, the question is again whether it is possible to educate all age groups on them. Is there such educational content that will develop the knowledge of the target group and can be shown on the selected geosite? In reality, there are very few locations that can appeal to both professionals and all other categories of target groups, including preschool children. Therefore, it is appropriate to talk about the educational potential rather than the educational value, which depends on the target group.

During the implementation of the experiment, a value of 70% was set as the "magical limit" of success. Why this number? Each group is composed of individuals who have different levels of knowledge, intellectual abilities, and motivation to participate in the educational program. Some authors then distinguish three levels of educational outputs, namely the minimal, optimal, and excellent levels [37,38]. The goal of the educational program should be for the participants to be able to successfully complete the inquiry-based activity at an optimal and excellent level. Since the distribution of these three levels (depending mainly on the IQ value) roughly corresponds to a Gaussian curve (16% minimal, 68% optimal, 16% excellent) [39], theoretically 84% of the participants should complete the activity. However, because some of them are less motivated and some make mistakes, the failure rate increases and is almost double under normal conditions. Therefore, the limit of 70% is more realistic. Nevertheless, it is more of an indicative figure.

Within the described method, a procedure is presented in which the educational program is first tested on a target group of a higher level, and only after a successful result is an attempt made to apply it to a lower level. The reason for this procedure is that geotourism educational programs currently do not have such a position in the offer of various leisure activities that it would be possible to make mistakes too often. When the participant of the program gets lost in a number of technical terms, or the topic of the excursion does not interest him, he will not come again next time. Similarly, if a program for a school misses its target group, the school will no longer order it. Therefore, it is necessary to create not only professionally processed but also professionally targeted programs. The selection of suitable geosites also belongs to this.

The author was motivated to write this article by the fact that a number of geosites with high scientific value have been identified as excellent locations from the point of view of education. At the same time, the reality was completely opposite. Many geosites were usable at most for university students of geology, but they were completely uninteresting to anyone else. Yes, it is also possible to introduce an excursion to these geosites and carry out an explanation or some form of activation educational method, but the resulting impression tends to be embarrassing. Different target groups prefer different kinds of geosites and expect different kinds of programs (Table 4). Small children are not so much interested in the aesthetic perception of the location; they are much happier when they can play in the given place in different ways. But even this game can educate them. Older children and ordinary tourists especially appreciate visually attractive geosites; as the level of knowledge increases, the importance of an interesting story told by the guide grows. At the same time, lower levels of education are certainly not less important. On the contrary, if education in a certain area is underestimated at a young age, it is difficult to make up for the deficit in the motivation of young people later.

How does one evaluate the result of the experiment when for category 3 the testing did not reach the expected success rate? Does this mean that the program needs to be

adjusted, but that the geosite continues to have the high educational potential for this category? Yes, it only means that the tested variant of the program, using a geosite and a certain topic at the selected level of difficulty, is not suitable for the given category. The question is, however, whether after adjusting the excursion in the selected route it still makes sense. If we take the example of Velký Jelení vrch, where a possible adjustment would be to simplify the message on the difference between sedimentary and volcanic rocks, visiting some geosites during the excursion is meaningless after this simplification. Likewise even a visit to Velký Jelení vrch itself, because in that case, the excursion could only lead to the first geosite (Děvín) and back. We would find everything we needed to see in this first section. In that case, however, it would be best to propose a completely different excursion. That is why the educational potential of Velký Jelení vrch for category 3 is significantly lower than for categories 4 and 5, and for categories 1 and 2, it is almost zero, as it is not possible to make a safe and interesting program for the given age category here due to the exposed summit crags.

## 5. Conclusions

The aim of this paper was to present a method of assessing the educational potential of a geosite, based on the evaluation of the success rate of the inquiry-based activity and the final test, verifying what the participants of the educational program have remembered. The method also uses a procedure where the selected educational program is first tested with a group that can be expected to have a deeper knowledge of the topic, and then, based on an evaluation of the success rate, it can be used with younger or less experienced groups. With this procedure, we try to ensure that the program is not too scientific, because in this case the participants often give up on trying to understand the guide's interpretation, start focusing on other things, and probably will not come to another similar program next time.

The article also discusses the relativity of the geosite's educational value. The author points to the fact that the educational potential always depends on the target group for which the prepared program is intended. Any educational program (in the case of geoparks, it is typically an excursion) must first of all be a great experience that leaves the participants with a good feeling, touches their emotions, and awakens in them the motivation to further educate themselves on the topic. When we want to create the best educational program for a certain target group, the selection of visited geosites must take into account their preferences, limits, knowledge, and abilities, etc. It is especially important to rightly choose the central geosite of the entire excursion, which should be an aesthetic highlight, and the story told by the guide should culminate here. Depending on what the story is, we then try to choose a suitable geosite.

The author of the article is aware of the great degree of subjectivity that is present in the given method. If the guide makes the final test unreasonably difficult, the pass rate will hardly be higher than the required 70%. If the guide does not explain certain information very well, the success rate will again be lower, even though the topic is reasonably expert for the target group. However, when an experienced guide starts using this methodology, he can create a database of combinations of geosites and topics in the territory in which he operates, forming an offer of educational programs for different target groups. Precisely targeted programs increase participants' sense of the guide's professionalism and spread the good name of geoparks. The database, in which suitable programs for selected target groups can be easily filtered, then facilitates the planning of orders. However, even if this method (successfully verified for the purposes of the Ralsko National Geopark) was not used in practice in other places, the author at least hopes that this article will spark a discussion about assessing the educational value of geosites, as he considers the attempt to numerically express some kind of "objective quality" already overcome.

**Funding:** This research received no external funding.

**Institutional Review Board Statement:** The authors declare that the study was conducted in accordance with the ethical rules that are generally accepted for humanities research.

**Informed Consent Statement:** Informed consent was obtained from all subjects involved in the study. All respondents participated in the research voluntarily and were informed about it in advance.

**Data Availability Statement:** Data are available on request from the author.

**Conflicts of Interest:** The authors declare no conflict of interest.

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
