# Peer review of "Assessing the Educational Potential of Geosites: Introducing a Method Using Inquiry-Based Learning"

_resources, doi:10.3390/resources11110101_

Round 1
Reviewer 1 Report
- In abstract, some concrete results of the study could be mentioned
- In key words: geographical location of study. I don't think it's necessary- georanger guiding
- There are numerous other works on the given topic (geoeducation, geo- interpretation).
- Figure 2- north orientation; geographical coordinates and relief units.
- Figure 3- Whats means CIL? Point 1?Scale? north?
- Figure 4- What means colours? Legend is missing
- the discussion part can be deepened
Author Response
Dear reviewer,
thank you for your comments. Based on your recommendations, I made the following changes to the article:
- I added a sentence to the abstract, specifically describing the evaluation method's principle.
- I changed the keywords as per your recommendation.
- Figure 2 - I added north arrow, but I didn't add relief units because the map would be too confusing (too much text).
- Figure 3 - I fixed the error with "CIL"
- Figure 4 - I have added a link to the legend, which I do not include in the article, as it is too extensive. However, the colors correspond to international standards.
Reviewer 2 Report
Excellent article, very current and important in relation to educational studies related to geosites and geodiversity.
I particularly missed more images, mainly field images showing the different types of rocks and associated geological features. In addition, field images of the different groups at the time of evaluation, for example.
Figure 4 is not a geological map in itself, as it lacks several important information, such as lithologies, structures and of course, not to mention the legend.
Author Response
Dear reviewer,
thank you for your comments. Based on your recommendation, I have added a figure showing the three basic rock types discussed in the article. Unfortunately, due to European legislation on the protection of personal rights (GDPR), I cannot insert pictures of the various evaluated groups into the article, as I would need the consent of all those involved (in the case of children of their parents), which I do not have. But I hope that this figure is not so important for the article.
I also corrected the title of Figure 4 and added a link to the legend (the legend is too extensive, that's why I don't mention it directly in the article).
Reviewer 3 Report
Dear Author,
Congratulation!!!
The necessity to adjust the level of knowledge transfer in the geospatial site to the recipient is crucial. Likewise, the type of content that is conveyed to him. A very important article that collects the advantages and disadvantages of knowledge transfer, which should be read by every geointerpreter who cares about the effectiveness of his argument.
Author Response
Dear reviewer, thank you for your kind evaluation of the article. I also hope that it will be inspirational to different people involved in geointerpretation.